A diagnostic model for minimal change disease based on biological parameters

Zhu Hanyu 1
Han Qiuxia 2
Zhang Dong dongzhang301301@126.com 1
Wang Yong wangyong301@263.net 1
Gao Jing 3
Geng Wenjia 4
Yang Xiaoli 1
Chen Xiangmei 1
1 Department of Nephrology, Chinese PLA General Hospital, Chinese PLA Institute of Nephrology, State Key Laboratory of Kidney Diseases, National Clinical Research Center of Kidney Diseases, Beijing Key Laboratory of Kidney Disease , Beijing , China
2 Department of Nephrology, The First Affiliated Hospital of Zhengzhou University , Zhengzhou , China
3 Department of Clinical Biochemistry, Chinese PLA General Hospital , Beijing , China
4 Department of Nephrology, Guangdong Provincial Hospital of Chinese Medicine , Guangzhou , China
Kennedy David
Electronic publication date: 2018 Jan 12
Publication date: 2018
Volume: 6
Electronic Location ID: e4237
Received 2017 Sep 29; Accepted 2017 Dec 15
Copyright: ©2018 Zhu et al.
Copyright year: 2018
Copyright holder: Zhu et al.
License: This is an open access article distributed under the terms of the Creative Commons Attribution License, which permits unrestricted use, distribution, reproduction and adaptation in any medium and for any purpose provided that it is properly attributed. For attribution, the original author(s), title, publication source (PeerJ) and either DOI or URL of the article must be cited.
License URL: https://creativecommons.org/licenses/by/4.0/

Keywords: Minimal change disease, Diagnostic model, Biological parameters

Funding: National Key R&D Program of China 2016YFC1305500 National Natural Science Foundation of China 61671479 61471399 81401719 Innovation Nursery Fund of PLA General Hospital 15KMZ04 This study was supported by the National Key R&D Program of China (2016YFC1305500); the National Natural Science Foundation of China (No. 61671479); the National Natural Science Foundation of China (No. 61471399); Innovation Nursery Fund of PLA General Hospital (No. 15KMZ04); and the National Natural Science Foundation of China (No. 81401719). The funders had no role in study design, data collection and analysis, decision to publish, or preparation of the manuscript.

==============================
Background

Minimal change disease (MCD) is a kind of nephrotic syndrome (NS). In this study, we aimed to establish a mathematical diagnostic model based on biological parameters to classify MCD.

Methods

A total of 798 NS patients were divided into MCD group and control group. The comparison of biological indicators between two groups were performed with t-tests. Logistic regression was used to establish the diagnostic model, and the diagnostic value of the model was estimated using receiver operating characteristic (ROC) analysis.

Results

Thirteen indicators including Anti-phospholipase A2 receptor (anti-PLA2R) (P = 0.000), Total protein (TP) (P = 0.000), Albumin (ALB) (P = 0.000), Direct bilirubin (DB) (P = 0.002), Creatinine (Cr) (P = 0.000), Total cholesterol (CH) (P = 0.000), Lactate dehydrogenase (LDH) (P = 0.007), High density lipoprotein cholesterol (HDL) (P = 0.000), Low density lipoprotein cholesterol (LDL) (P = 0.000), Thrombin time (TT) (P = 0.000), Plasma fibrinogen (FIB) (P = 0.000), Immunoglobulin A (IgA) (P = 0.008) and Complement 3 (C3) (P = 0.019) were significantly correlated with MCD. Furthermore, the area under ROC curves of CH, HDL, LDL, TT and FIB were more than 0.70. Logistic analysis demonstrated that CH and TT were risk factors for MCD. According to the ROC of “CH+TT”, the AUC was 0.827, with the sensitivity of 83.0% and the specificity of 69.8% (P = 0.000).

Conclusion

The established diagnostic model with CH and TT could be used for classified diagnosis of MCD.

Introduction

Minimal change disease (MCD) is a common pathological type of nephrotic syndrome (NS), and its typical characteristic is the diffuse effacement of podocyte foot processes observed by electron microscope (Glick, 2007). The actin of foot process is linked with slit diaphragm, which is important for renal glomerular filtration function. MCD is commonly seen in child patients with NS. According to the reports, about 80% MCD cases are aged less than 10 years old. The incidence of MCD in adults is lower than that in children (Cameron, 1996; Haas et al., 1997; Kazi et al., 2009; Zech et al., 1982). In our country, the incidence of adult MCD patients in NS patients is less than 25% (Chu, Chen & Liu, 2014; Zhou et al., 2011). At present, renal biopsy, which is an invasive examination, is required for most glomerulopathy diagnoses. Although it can offer the diagnosis and testing information for the doctors, renal biopsy may cause complications, such as bleeding. Moreover, some patients can not accept the renal biopsy, leading to the absence of timely diagnosis (Fiorentino et al., 2016; Magistroni et al., 2015; Verde et al., 2012). Therefore, a non-invasive model is urgently needed to discriminate MCD.

It is reported that mathematical model, like classification and regression tree (CART) model can be used as a method to classify different diseases (Hu et al., 2011b; Yan, Lin & Liu, 2011). At present time, the diagnostic model based on data analysis has become the focus of disease diagnosis, and it also can be used in the noninvasive diagnosis (Azmak et al., 2015). Moreover, some reports have shown that the classification equations have been used in kidney diseases.

In present study, with the purpose of classifying MCD and other kidney diseases, we established a diagnostic model based on the clinical parameter. Additionally, we also conducted common statistical analyses, including Chi-square tests, logistic analysis and receiver operating characteristic (ROC) analysis.

Methods

Study object

This study was approved by the Medical Ethics Committee of the Chinese PLA General Hospital, and written consents were obtained from all patients. The inclusion criteria of the current research were listed as follows: (1) all the participants were the first time to be admitted into the Department of Nephrology of our hospital; (2) adult patients; (3) no one accepted the renal biopsy before entering our hospital; (4) no one accepted any treatments, including hypertension treatment or hyperlipidemia treatment; (5) no one suffered from any tumors, except hypertension, diabetes, hepatitis or lupus erythematosus; (6) all patients accepted renal biopsy during their hospitalization. The following exclusion criteria were applied in our study: (1) the patients could not accept renal biopsy; (2) no complete clinical data were provided. According to the inclusion and exclusion criteria, 798 patients were finally recruited, containing 47 MCD patients and 751 patients with other kidney diseases.

Samples and biological parameters

For all the 798 patients, their blood samples were collected on the second day after entering hospital. Then, blood coagulation test, blood routine examination and clinical biochemistry testing were performed. The demographic data as well as clinical and laboratory examination of all patients were recorded, including age, gender, presence of other diseases, physical examination, and so on.

Statistical analysis

In this study, all statistical analyses were performed using SPSS 19.0 and GraphPad Prism 5. The data were summarized and presented as means ± SD. The biological indicators of the two groups were assessed by using t-tests. Logistic regression was employed to establish the diagnostic model. The diagnostic value of the constructed model was examined via performing ROC analysis. P values less than 0.05 were considered to be statistically significant in this paper.

Results

The characteristics of patients

The tested biological parameters were all listed in Table 1, including Anti-phospholipase A2 receptor (anti-PLA2R), Alanine aminotransferase (ALT), Aspartate aminotransferase (AST), Total protein (TP), Albumin (ALB), Total bilirubin (TB), Direct bilirubin (DB), Alkaline phosphatase (ALP), γ-Glutamyltransferase (GGT), Glucose (GLU), Urea nitrogen (UN), Creatinine (Cr), Uric acid (Ua), Total cholesterol (CH), Triglyceride (TG), Creatine kinase (CK), Lactate dehydrogenase (LDH), High density lipoprotein cholesterol (HDL), Low density lipoprotein cholesterol (LDL), Thrombin time (TT), Prothrombin time (PT), Plasma fibrinogen (FIB), D-dimer (D2), Immunoglobulin A (IgA), Immunoglobulin G (IgG), Immunoglobulin M (IgM), Immunoglobulin E (IgE), Complement 3 (C3), Complement 4 (C4) and Body mass index (BMI). Moreover, the reference ranges of them were also listed in the table. The demographic data and history of diseases of these two groups were recorded in Table 2. We found that the rates of hypertension and diabetes were declined in MCD patients compared with the patients with other kidney diseases. In MCD group, the numbers of patients less than 40 years old and more than 40 years old were about the same, and the similar result was found in the group of other kidney diseases. We observed more male patients than female patients in both groups. Besides, most patients of the two groups had no hypertension, diabetes or hepatitis. The BMI value was 25.03 ± 4.66 in MCD group, and the data for the group of other kidney disease was 25.45 ± 4.37.

Table 1 The biological parameters in this study.

The tested biological parameters.

Index full name	Abbreviation	Reference range	
Anti-phospholipase A2 receptor	Anti-PLA2R		
Alanine aminotransferase	ALT	0–40 U/L	
Aspartate aminotransferase	AST	0–40 U/L	
Total protein	TP	55–80 g/L	
Albumin	ALB	35–50 g/L	
Total bilirubin	TB	0–21 µmol/L	
Direct bilirubin	DB	0–8.6 µmol/L	
Alkaline phosphatase	ALP	0–130 U/L	
γ-Glutamyltransferase	GGT	0–50 U/L	
Glucose	GLU	3.4–6.2 mmol/L	
Urea nitrogen	UN	1.8–7.5 mmol/L	
Creatinine	Cr	30–110 µmol/L	
Uric acid	Ua	104–444 µmol/L	
Total cholesterol	CH	3.1–5.7 mmol/L	
Triglyceride	TG	0.4–1.7 mmol/L	
Creatine kinase	CK	2–200 U/L	
Lactate dehydrogenase	LDH	40–250 U/L	
High density lipoprotein cholesterol	HDL	1–1.6 mmol/L	
Low density lipoprotein cholesterol	LDL	0–3.4 mmol/L	
Thrombin time	TT	16.0–18.0 s	
Prothrombin time	PT	11.0–15.0 s	
Plasma fibrinogen	FIB	200–400 mg/dL	
D-dimer	D2	0.0–0.5  µg/L	
Immunoglobulin A	IgA	70–180 mg/dl	
Immunoglobulin G	IgG	700–1,600 mg/dl	
Immunoglobulin M	IgM	40–230 mg/dl	
Immunoglobulin E	IgE	0–100 IU/ml	
Complement 3	C3	90–180 mg/dl	
Complement 4	C4	10–40 mg/dl	
Body mass index	BMI	18.5–24.99	

Table 2 Basic information of the two groups.

The demographic data and history of diseases of these two groups.

	MCD group (n = 47)	Group of other kidney diseases (n = 751)	P value	
Age			0.452	
<40	25	357	
≥40	22	394	
Gender			0.685	
Male	30	457	
Female	17	294	
Hypertension			0.000	
Yes	7	363	
No	40	388	
Diabetes			0.041	
Yes	2	113	
No	45	638	
Hepatitis			0.050	
Yes	0	57	
No	47	694	
BMI	25.03 ± 4.66	25.45 ± 4.37	0.895	

The comparison of biochemical indicators between two groups

In order to explore the association between biochemical indicators and MCD, student’s t-test was performed. As shown in Table 3, the results showed that among the 28 biochemical indicators, 13 indicators including anti-PLA2R (P = 0.000), TP (P = 0.000), ALB (P = 0.000), DB (P = 0.002), Cr (P = 0.000), CH (P = 0.000), LDH (P = 0.007), HDL (P = 0.000), LDL (P = 0.000), TT (P = 0.000), FIB (P = 0.000), IgA (P = 0.008) and C3 (P = 0.019) were significantly different between the two groups.

Table 3 The comparison of serological parameters in the two groups.

Parameter	MCD group (n = 47) (mean ± SD)	Group of other kidney diseases (n = 751) (mean ± SD)	P value	
Anti-PLA2R	2.00 ± 0.02	34.22 ± 114.96	0.000*	
ALT	27.00 ± 35.12	22.61 ± 21.99	0.201	
AST	20.79 ± 8.80	18.82 ± 11.39	0.240	
TP	43.32 ± 10.01	58.33 ± 12.02	0.000*	
ALB	22.68 ± 7.87	33.94 ± 9.04	0.000*	
TB	7.87 ± 3.64	8.63 ± 4.40	0.177	
DB	1.39 ± 0.99	1.98 ± 1.32	0.002*	
ALP	69.36 ± 34.22	66.03 ± 31.66	0.835	
GGT	41.43 ± 82.16	34.00 ± 51.21	0.398	
GLU	4.62 ± 0.88	4.98 ± 1.75	0.102	
UN	6.78 ± 4.30	6.30 ± 3.59	0.446	
Cr	82.25 ± 23.15	104.31 ± 68.19	0.000*	
Ua	345.71 ± 104.69	366.78 ± 102.00	0.198	
CH	8.81 ± 3.23	5.72 ± 2.21	0.000*	
TG	2.35 ± 1.60	2.22 ± 1.56	0.582	
CK	97.12 ± 88.43	95.70 ± 99.36	0.756	
LDH	204.66 ± 58.78	182.88 ± 59.31	0.007*	
HDL	1.74 ± 0.76	1.26 ± 0.84	0.000*	
LDL	6.31 ± 2.81	3.81 ± 1.84	0.000*	
TT	18.65 ± 2.52	16.56 ± 1.44	0.000*	
PT	12.85 ± 0.83	13.15 ± 1.55	0.243	
FIB	5.46 ± 1.79	4.22 ± 1.50	0.000*	
D2	1.07 ± 0.79	0.98 ± 1.84	0.721	
IgA	220.47 ± 92.70	259.46 ± 117.18	0.008*	
IgM	142.22 ± 64.44	111.43 ± 126.53	0.099	
IgE	445.22 ± 927.74	205.44 ± 894.09	0.090	
C3	116.86 ± 27.12	107.55 ± 26.38	0.019*	
C4	28.15 ± 10.12	26.46 ± 9.49	0.238	
Notes.

* P < 0.05.

ROC analysis of related characteristics

The ROC analysis was conducted to detect the diagnostic value of these 13 indicators, and the results were displayed in Fig. 1. We found that in Fig. 1, the area under the curves (AUCs) of CH, HDL, LDL, TT and FIB were more than 0.70, and the AUCs of them were 0.807, 0.746, 0.776, 0.817 and 0.713, respectively (P = 0.032, 0.037, 0.039, 0.032, and 0.046, respectively).

Figure 1 The ROC curves of anti-PLA2R, TP, ALB, DB, Cr, CH, LDH, HDL, LDL, TT, FIB, IgA and C3, the related indicators of MCD.

The ROC analysis was conducted to detect the diagnostic value of these 13 indicators.

Logistic analysis of the pre-selected parameters

In order to establish the classification models of MCD and other kidney diseases, the logistic analysis was carried out. From Table 4, we could see that CH and TT were risk factors for MCD, and the P values of them were both 0.000. Furthermore, the classification equation including CH and TT was as follows: PRE=1∕1+e−10.617−0.270×CH−0.325×TT.

Then, based on logistic regression of the predicted probability (PRE), the ROC curve of “CH+TT” is presented in Fig. 2. From Fig. 2, we could see that the AUC of “CH+TT” was 0.827, with the sensitivity of 83.0% and the specificity of 69.8% (P = 0.000).

Figure 2 The ROC curve of “CH+TT” combination from logistic regression of the predicted probability for MCD patients.

Discussion

MCD is a kind of glomerular disease caused by lesions of the podocyte. Most MCDs are idiopathic nephrotic syndromes among children and adults. MCD is characterized by hypoalbuminemia, hyperlipidemia, proteinuria and edema (Braden et al., 2000; McGrogan, Franssen & De Vries, 2011). It has been reported that the outcome of MCD is correlated with some elements, such as virus infection, drugs, allergy and even tumors (Korzets et al., 1992; Meyrier et al., 1992). At present, the pathogenesis of MCD still remains unclear, but many scholars consider that it may be associated with podocyte injury. Now in clinic, renal biopsy centesis is still the golden standard for the diagnosis of nephropathy (Appel & Appel, 2009; Floege & Eitner, 2011). Renal biopsy centesis is safe and easy operation, but it is invasive with risks (Fisi et al., 2012). Because some patients have suffered from absolute and relative contraindications of renal biopsy (Mohamed & John, 2011; Whittier & Korbet, 2004), they refuse to undergo renal biopsy. Since the conditions of hospital are not adequate, renal biopsy can not be implemented in every patient who is suspected to have kidney diseases. Moreover, the histopathological diagnosis of renal biopsy is not stable (Lu et al., 2011). Consequently, the non-invasive methods with high accuracy are urgently needed for MCD diagnosis are needed.

The mathematical model based on the statistical analysis and computer technique has been used in clinic, such as computed tomography (CT) and nuclear magnetic resonance (NMR) (Bandak et al., 1995; Hu et al., 2011a). Levey et al. (2009) established a new equation of estimated glomerular filtration rate (eGFR), named CKD-EPI, which could assess the stages of kidney diseases (CKD stage) (Levey et al., 2009). Gao et al. (2011) also established a diagnostic model for IgA nephropathy with 6 serum biochemical parameters, including ALB, CK, Cr, HDL, CA125 and TB (Gao et al., 2011). Additionally, they certified that this model could be used in the auxiliary diagnosis of IgA nephropathy.

In our study, a diagnostic model was established for MCD, and the enrolled patients were divided into two groups, which were MCD group and group of other kidney diseases. The present paper enrolled 47 MCD patients and 751 patients with other kidney diseases. In the two groups, the numbers of patients older than 40 years old and younger than 40 years old were almost equal. The ratio of male patients and female patients was about 2:1 in both groups, and very few patients suffered from hypertension, diabetes or hepatitis. Besides, the BMI of the two groups also had no significant differences. The t-test was implemented to analyze the serum biological indicators, and anti-PLA2R, TP, ALB, DB, Cr, CH, LDH, HDL, LDL, TT, FIB, IgA and C3 were significantly different between the two groups. Like the previous study by Gao and his colleagues, which also found 15 significant different serological indicators between IgAN patients and non-IgAN patients, the ROC analysis was performed to further assess the diagnostic value of the collected parameters (Gao et al., 2012). In our study, ROC analysis for the 13 indicators revealed that the AUCs of five indicators, CH, HDL, LDL, TT and FIB were more than 0.70. Logistic analysis was performed with CH, HDL, LDL, TT and FIB. The results suggested that CH and TT were risk factors for MCD. Based on logistic regression of the predicted probability on the two indicators, the results of ROC curve revealed that “CH +TT” the AUC of “CH +TT” was 0.827, with the sensitivity of 83.0% and the specificity of 69.8%.

Table 4 The multivariate logistic regression analysis for the model.

	B	S.E.	Wald	df	Sig.	Exp (B)	95% CI	
							Lower	Upper	
CH	−.270	.058	21.720	1	.000	.763	.682	.855	
HDL	−.186	.099	3.516	1	.061	.830	.684	1.008	
TT	−.325	.087	14.145	1	.000	.722	.610	.856	
Constant	10.617	1.449	53.669	1	.000	40,819.656			

In this study, we found that the combined diagnostic value of “CH +TT” was significantly higher than either of them alone. The combined diagnostic model might help improve the diagnosis of MCD, especially for those challenging case. However, the present study has some limitations. Firstly, the sample size of MCD group is not large enough. Moreover, all the patients were collected from a single institution and population, which might cause bias to the final results. Secondly, the specificity was not high in our study, leading to high false positive rate. The diagnostic accuracy of the combined model was 70.4%. In order to improve the diagnosis specificity, we could prioritize diagnosis sensitivity based on the trade-off between diagnosis false positive cases versus omitting true positive cases. According to ROC analysis, the diagnostic specificity was adjusted as 70%, and the sensitivity was 80.9%, while the diagnostic accuracy was 70%, which also hold great potential for clinical application. Thirdly, the diagnostic performance of the constructed models was only verified in the populations collected in the original analysis. A cross-validation was not set to investigate the diagnostic performance of the combined model for MCD in clinic. Additionally, all the MCD cases collected in our study were adults. However, MCD is frequently diagnosed among children, and the adult cases only account for about 10%–15%. The main reasons for childhood MCD include congenital anomalies and inherited disorders, but the diagnostic value of the combined model for the childhood MCD cases remained unknown (Downie et al., 2017; Ingelfinger, Kalantar-Zadeh & Schaefer, 2016). The distinct etiologies may lead to various clinical symptoms, biological parameters, and therapeutic responses. Thus, further investigations are needed to determinate whether the selected biological parameters exerted advantages in diagnosing MCD among children. Further related research will be carried out to address the above issues.

Conclusion

This study has established a diagnostic model based on the clinical parameters to classify MCD and other kidney diseases. The combined diagnostic model with CH and TT could effectively distinguish MCD from other nephrotic syndrome.

Supplemental Information

Supplemental Information 1 ROC-1 raw data

The data used in ROC-1 (Fig. 1).

Click here for additional data file.

Supplemental Information 2 ROC-2 raw data

Raw data used in ROC-2 (Fig. 2).

Click here for additional data file.

Supplemental Information 3 Table 3 raw data

Click here for additional data file.

Supplemental Information 4 Logistic raw data

Raw data used in the logistic analysis.

Click here for additional data file.

Additional Information and Declarations

Competing Interests

Author Contributions

Data Availability

The authors declare there are no competing interests.

Hanyu Zhu and Qiuxia Han performed the experiments, analyzed the data, contributed reagents/materials/analysis tools, wrote the paper, prepared figures and/or tables, reviewed drafts of the paper.

Dong Zhang and Yong Wang conceived and designed the experiments.

Jing Gao analyzed the data.

Wenjia Geng and Xiaoli Yang contributed reagents/materials/analysis tools, wrote the paper.

Xiangmei Chen reviewed drafts of the paper.

The following information was supplied regarding data availability:

The code is included in the Results section of the manuscript.

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
