# Peer review of "A diagnostic model for minimal change disease based on biological parameters"

_PeerJ, doi:10.7717/peerj.4237_

## Round 0.1 · original submission · Major Revisions

While you should respond in a specific and detailed manner to each point raised by the reviewers, please make particular note of additional methodological clarification/analysis needed in the statistical analysis, particularly that pertaining to the ROC analysis which was a concern noted by both reviewers.

Reviewer 1 ·

Basic reporting

While manuscript is written in easy-to-read English language, it can be made more structured and of professional quality by substantially improving syntax errors, e.g....

* Excessive use of abbreviations. Group them when applicable in abstract and under results section, e.g., type of proteins measured, immunoglobulins, etc. and suggest brief explanation on their relevance in MCD.
* Provide specific examples instead of using keywords, e.g., "certain risk" (line 34), "widely used" or "mathematical model" (line 35), etc.

Experimental design

1. My first major issue is with rationale (line 56-66) for development of mathematical model which is to avert invasive nature of the renal biopsy in patients with NS. Definitive diagnosis of MCD requires renal biopsy but it warrants electron microscopic examination as renal parenchyma looks almost normal under light microscopy. However, patients, mostly children with NS secondary to MCD have very typical clinical presentation with marked dependent edema, weight gain with growth retardation along with marked albumin secretion in urine and decrease in serum albumin. Unlike other causes of NS, MCD disease have usually normal renal function and lacks autoimmune dysfunction. Hence, present clinical guidelines rarely requires renal biopsy for diagnostic purpose and biopsy is usually reserved for rare cases (1-5%) where patient do not respond to steroids (majority does) or MCD is seen in adults (rarely seen). Since authors aim to provide their method for diagnostic purpose, it would help to have strong rationale to promote its clinical application. Such non-invasive approach, if accurate enough can be of much value for smaller fraction (< 1-5%) of cases where diagnosis is challenging, e.g., adult cases with hypertension, autoimmune disorders, etc. However, for larger cohort of patients of MCD are children (80-90%) and I have concerns and seek authors view on how their method provide additional value to existing clinical guidelines.

2. Another important issue is with methodology for developing risk prediction model. While t-test and logistic regression based model should work in MCD given a typical (a few features with less variance) patient presentation as described earlier, most biochemical parameters that Zhu et al. used (table 1) are part of standard or extended panel of investigations carried out by a physician for a patient presenting with NS. In that case, it would be useful to see how authors' model perform with and without including baseline biochemical indicators, e.g., total protein, albumin, creatinine, and lipid profile. Also, unlike reported ROC statistics in training set of 798 cases, it is equally important to report ROC statistics in cross-validation set (split from training set of 798 cases) if not the gold-standard, test or held-out set which is independent from the training set.

3. Under ROC statistics results, it is useful to provide ROC statistics for combined model using all 13 parameters and perform stepwise forward or backward selection in estimating coefficients of each of 13 parameters to predict which of parameters are of high vs low predictive value.


References:

1. Kumar, Abbas, Aster. Robbins Pathologic Basis of Disease. 9th edition (2015)
2. Mansur & Batuman. Minimal-Change Disease. 6/30/2017 http://emedicine.medscape.com/article/243348-overview

Validity of the findings

1. With biomarker based prediction models, it is reasonable to not expect high level of accuracy and hence, authors reporting of sensitivity and specificity in range of 70-80% seems fair. However, I suggest authors can improve discussion section by adding a note on prioritizing sensitivity or specificity depending of trade-off between diagnosis false positive cases versus omitting true positive cases, respectively.

2. In discussion section, authors show total cholesterol and thrombin time as risk factors with highest predicted probability. Although these are known risk factors and often used in secondary panel of investigation to evaluate patients with NS, it would be useful to know if predicted probability of these markers remains high also among cases (from their training set) where diagnosis of MCD was challenging. If so, this can be an important finding to let physician weigh more on values of these two indices over others for the challenging case.

Additional comments

Zhu et al. have presented a logistic regression based diagnostic approach for minimal change disease (MCD), a common pathological feature of nephrotic syndrome (NS) in children. Authors have provided detailed supplemental data and description on their rationale, statistical methods and visualization of prediction results using standard ROC plots. After reading the manuscript in totality, I find following several issues (under three sections) which I suggest authors to address before it is being considered for second review.

Reviewer 2 ·

Basic reporting

The article is clearly presented

Experimental design

the article is mathematical model for kidney disease in particular Minimal Change Disease. It is retrospective in nature. the data set is appropriate

Validity of the findings

The validity of the findings may not transcend to other ethnic groups
The authors note that HTN and diabetes were elevated in the MCD group. this does not appear to be the case in the table.
It the best of my reading the ROC was not applied to a subset of individuals not included in the original analysis
It is not clear if individuals entering the study were on other treatment for example therapy for HTN or hyperlipidemia

Additional comments

I believe that the study is very interesting but would find it hard to generalize this to other population subsets

---

## Round 0.2 · accepted · Accept

You have adequately satisfied both Reviewers concerns and we look forward to disseminating this interesting work.

Reviewer 1 ·

Basic reporting

None

Experimental design

None

Validity of the findings

None

Additional comments

I thank authors for having detailed look on their manuscript and addressing most of my comments. With respect to experimental design, given case population is all adults and issues with availability of renal biopsy setup in hospitals, it is reasonable to have computational based predictions to provide some level of support in addition to biochemical indices in cases with diagnostic dilemma. Accordingly, prediction model by authors may complement current diagnostic approach for MCD in adults. As prediction models are inherently prone for overfitting and resulting inaccuracy, I wish authors to follow up with at least cross-validated model if not the model performing better (arbitrary value of AUC ~>0.75 or more) based on actual validation set. Such statistical rigor on top of their approach to include biochemical indices in their existing prediction model should yield reliable non-invasive approach for adult patients with MCD.

Reviewer 2 ·

Basic reporting

appropriate

Experimental design

the authors have addressed concerns

Validity of the findings

the authors have addressed concerns

Additional comments

the authors have addressed concerns